# Potential Protective Effects of Equol (Soy Isoflavone Metabolite) on Coronary Heart Diseases—From Molecular Mechanisms to Studies in Humans

**DOI:** 10.3390/nu13113739

**Published:** 2021-10-23

**Authors:** Xiao Zhang, Cole V. Veliky, Rahel L. Birru, Emma Barinas-Mitchell, Jared W. Magnani, Akira Sekikawa

**Affiliations:** 1Department of Epidemiology, Graduate School of Public Health, University of Pittsburgh, Pittsburgh, PA 15213, USA; xiz186@pitt.edu (X.Z.); cvv2@pitt.edu (C.V.V.); rlb63@pitt.edu (R.L.B.); ejb4@pitt.edu (E.B.-M.); 2Department of Medicine, University of Pittsburgh, Pittsburgh, PA 15213, USA; magnanij@pitt.edu

**Keywords:** equol, isoflavones, flavonoid, lipophilicity, inflammation, oxidation, endothelial function, arterial stiffness, atherosclerosis, coronary heart disease

## Abstract

Equol, a soy isoflavone-derived metabolite of the gut microbiome, may be the key cardioprotective component of soy isoflavones. Systematic reviews have reported that soy isoflavones have no to very small effects on traditional cardiovascular disease risk factors. However, the potential mechanistic mode of action of equol on non-traditional cardiovascular risk factors has not been systematically reviewed. We searched the PubMed through to July 2021 by using terms for equol and each of the following markers: inflammation, oxidation, endothelial function, vasodilation, atherosclerosis, arterial stiffness, and coronary heart disease. Of the 231 records identified, 69 articles met the inclusion criteria and were summarized. Our review suggests that equol is more lipophilic, bioavailable, and generally more potent compared to soy isoflavones. Cell culture, animal, and human studies show that equol possesses antioxidative, anti-inflammatory, and vasodilatory properties and improves arterial stiffness and atherosclerosis. Many of these actions are mediated through the estrogen receptor β. Overall, equol may have a greater cardioprotective benefit than soy isoflavones. Clinical studies of equol are warranted because equol is available as a dietary supplement.

## 1. Introduction

Coronary heart disease (CHD) is the leading cause of morbidity and mortality worldwide [1]. Soy is a potential nutritional source for preventing CHD [2] and is a standard part of a traditional diet in East Asia [3]. The main components of soy that may exert cardioprotective effects are soy isoflavones (ISFs), mainly daidzein and genistein [4]. ISFs are phytoestrogens and are structurally similar to estradiol [5]. Estradiol exerts its biological action by binding both the estrogen receptor α (ERα), expressed in the reproductive, central nervous, and cardiovascular systems, as well as other systems [6], and the estrogen receptor β (ERβ), expressed in the cardiovascular and central nervous systems, as well as other systems. ISFs, however, preferentially bind to ERβ [7]. Among the flavonoid family of flavones, flavonols, flavanones, and ISFs, ISFs have the highest lipophilicity, which could contribute to their greater absorption from the gut [8]. ISFs may lower the risk of CHD by reducing inflammation [9,10,11] and oxidative stress [12,13,14]; the latter may prevent the oxidative damage to low-density lipoprotein (LDL) that contributes to atherogenesis [15].

Although there are clear cardiovascular benefits of ISFs in preclinical studies [16,17,18,19], the evidence in humans is conflicting [20,21,22]. A recent meta-analysis of 23 prospective cohort studies (330,826 participants) showed that the dietary intake of ISFs was inversely associated with all-cause and cancer mortality and yet not with cardiovascular disease (CVD) mortality [22]. Furthermore, ISFs have very small effects on the traditional CVD risk factors [23,24,25,26].

A growing hypothesis is that the ability of humans to metabolize daidzein to equol may contribute to the cardioprotective effects of ISFs [27,28,29]. A case-control study of myocardial infarction (MI), nested within a prospective cohort study, in women in China reported a significant inverse association of incident MI with urinary equol rather than ISFs and their other metabolites [30]. Cell culture and preclinical studies show that equol has a greater affinity for ERβ than its precursor daidzein, a longer half-life, greater bioavailability than daidzein and genistein, and a more potent antioxidant activity than any other ISFs [7,27]. Therefore, equol may be more cardioprotective than ISFs [27,29].

The mechanistic mode of action of equol is not completely understood. Investigation of the effects of equol has mostly been conducted in the in vitro assays and preclinical studies, while several studies in humans explored whether equol production caused the cardioprotective effects previously attributed to ISFs. Therefore, the aim of this systematic review is to summarize the current knowledge about the mechanisms underlying the potential protective effects of equol on atherosclerotic diseases. This review outlines evidence from in vivo, in vitro, and human studies on the anti-inflammatory, antioxidative, vasodilatory, and other effects of equol, as well as the association of equol with atherosclerosis, arterial stiffness, and CHD.

## 2. Materials and Methods

The PubMed database was searched from inception through to 28 July 2021 by two investigators (X.Z. and C.V.). Discrepancies were adjudicated with another investigator (A.S.). The MeSH terms used for the search were: equol, inflammation, oxidation, endothelial function, vasodilation, atherosclerosis, arterial stiffness, and CHD. We focused on these terms rather than the traditional CHD risk factors because the meta-analyses of human randomized controlled trials (RCTs) showed that ISFs had very small effects, if any, on the traditional cardiovascular risk factors [31,32]. A two-stage screening process consisting of a title-and-abstract scan and a full-text review was used to determine the eligibility of the articles. The exclusion criteria were: 1. Studies where equol was not reported; 2. human studies that were conducted among equol producers only, instead of between equol producers and non-producers, thus having a lack of contrast of the amount of equol; 3. studies where the equol was derived from plants other than soy, or the source of equol was unknown; 4. studies where the outcomes of the participants were not related to inflammation, oxidation, endothelial function, arterial stiffness, atherosclerosis, or CHD; 5. review articles or proposals; 6. studies that were published in a language other than English.

## 3. Results

The screening process is shown in Figure 1. Of the 231 articles identified, 69 met our criteria for review and were summarized.

### 3.1. Characterization of Equol

Equol is a compound that can exist as two isomers, S-equol and R-equol. However, only S-equol is produced in humans and animals. Thus, in this article, we use the term equol to denote S-equol only. Equol is the bioactive metabolite that is metabolized by intestinal bacteria from daidzein (Figure 2), and it has been suggested that it contributes to the health benefits of ISFs [33,34,35,36,37]. There is interindividual variation in the ability to produce equol [38,39]. Although the majority of animals produce equol, 30–70% of the adult human population can produce equol following soy challenge [38], and this capacity is due to the presence of specific gut microflora [40]. Some of the identified equol-producing bacteria are *Adlercreutzia equolifaciens*, *Asaccharobacter celatus*, *Enterorhabdus mucosicola*, *Slackia isoflavoniconvertens*, and *Slackia equolifaciens* [29,41,42,43]. The bacterial biosynthesis of equol from daidzein proceeds via a series of reduction reactions catalyzed by several reductases, such as nicotinamide adenine dinucleotide phosphate reductase, dihydrodaidzein reductase, and tetrahydrodaidzein reductase [44,45,46]. If an individual’s ability to produce equol is reported to be stable over 1 to 3 years [47], they can be classified as either “equol producers” or “non-producers”.

Equol has the following characteristics that may grant it with possibly greater cardioprotective properties than ISFs: (1) Equol has higher antioxidant activity than ISFs [33,39,51], and its antioxidant properties are even greater than vitamins C and E in in vitro studies [52]; (2) the affinity of equol to ERβ is much higher than daidzein, though it is similar to genistein [53]; (3) equol is more lipophilic than ISFs; therefore, it has a higher absorption rate from the gut than ISFs, resulting in greater bioavailability [8]; (4) equol has a longer plasma half-life than ISFs, being retained in the body for a greater period following soy consumption [39]; (5) equol has a lower ability to bind to serum proteins (i.e., albumin, sex hormone-binding globulin, and alpha-fetoprotein) than ISFs and thus has a greater availability for receptor occupancy [27].

### 3.2. ERβ

Estrogen receptors (ERs) are intracellular transcription factors whose activity is modulated by estrogens, non-steroidal estrogen antagonists, or agonists such as ISFs. ERs belong to the nuclear hormone receptor family with two subtypes, ERα and ERβ [6]. Accumulating evidence has suggested that ERβ may play a more dominant role in mediating the cardioprotective effects of estradiol [6]. ERβ knock-out mice have an increased mortality and exacerbated heart failure after MI [54]. Studies of ERβ knock-out mice exhibited poor functional recovery compared to the wild-type or ERα knock-out mice in an ex vivo model of global ischemia-reperfusion [55]. Consistently, cardiomyocyte-specific ERβ overexpression improved cardiac function and survival after MI induced by left anterior descending coronary artery ligation [6].

Equol is a selective ERβ agonist [7]. Studies demonstrated that ERβ agonists are cardioprotective. For example, one study demonstrated that a selective ERβ agonist led to a significant recovery of cardiac functionality in ovariectomized mice due to the ERβ-dependent upregulation of cardioprotective genes [56]. Moreover, the exposure of human macrophages to an ERβ-selective agonist led to a robust decrease in the level of extracellular heat shock protein 27, a biomarker of atherosclerosis [57], while the ERα-selective agonists did not show such an effect [58]. Subjects receiving ERβ-selective agonists may retain the benefits of hormone therapy without the severe adverse effects.

### 3.3. Potential Cardioprotective Properties of Equol

#### 3.3.1. Anti-Inflammatory Effects

Inflammation contributes to atherosclerosis from its inception to its development into thrombotic complications [59,60]. The accumulation of oxidized lipids triggers atherosclerosis mediated by proinflammatory cytokines, such as interleukin-1β, tumor necrosis factor-α (TNF-α), and nuclear factor-κB (NF-κB). Proinflammatory cytokines initiate the expression of vascular cell adhesion molecule-1 (VCAM-1) by endothelial cells (ECs) and allow the attachment of leukocytes, monocytes, and T lymphocytes to the arterial wall. Once adhered to the arterial endothelium, the monocytes penetrate the endothelial lining and enter the intima of the vessel wall by diapedesis between the ECs, a process triggered by chemokine monocyte chemoattractant protein-1 (MCP-1) [60]. Within the intima, monocytes mature into macrophages, exhibit increased expression of scavenger receptors, and engulf modified lipoproteins. Cholesterol esters accumulate in the cytoplasm, and the macrophages become foam cells [59,60].

Equol inhibits the overproduction of inflammatory biomarkers expressed by macrophages, microglia cells, and adipose tissues. In addition, equol ameliorates the inflammatory processes important to the pathogenesis of rheumatoid arthritis, metabolic syndrome, and intracranial aneurysm (Table 1, Figure 3).

**Table 1 nutrients-13-03739-t001:** Anti-inflammatory effects of equol.

#	Authors	Type	Findings	Has Effect
1	Blay [61]	In vitro	Equol (10 μM) significantly inhibited the overproduction of NO and PGE2 induced by LPS plus INF-γ when a pre-treatment was performed or when administered during activation. Moreover, equol regulated the gene transcription of cytokines and inflammatory markers. Genistein (20 μM) exerted similar anti-inflammatory effects, but daidzein did not.	Yes
2	Johnson [62]	In vitro	Equol exhibited protective effects against pro-inflammatory cytokines (IL-6 and TNF-α) and NO production in murine microglia cells. Equol also showed greater permeability through artificial gut and blood-brain barriers compared to daidzein.	Yes
3	Obiorah [63]	In vitro	Equol and ISFs induced endoplasmic reticulum stress and inflammatory response stress-related genes in a comparable manner to estrogens. Equol and ISFs induced proliferation of estrogenized breast cancer cells (simulating a perimenopausal state) but induced apoptosis of estrogen-deprived cells (simulating a postmenopausal state).	Yes
4	Nagarajan [64]	In vitro	In an in vitro LPS-induced inflammation model, equol dose-dependently inhibited LPS-induced MCP-1 secretion by macrophages.	Yes
5	Subedi [65]	In vitro	In microglial cells, equol inhibited TLR4 activation, MAPK activation, NF-kB-mediated transcription of inflammatory mediators, production of NO, release of PGE-2, and secretion of TNF-α and IL-6 in LPS-activated murine microglia cells.	Yes
6	Moriyama [66]	In vitro	Equol attenuated LPS-induced NO production with a concomitant decrease in the expression of iNOS. Equol did not affect the LPS-induced increase in intracellular ROS production. Increased NO production is a well-known inflammatory change in astrocytes stimulated by LPS. Attenuation of NO production by equol may mitigate LPS-induced neuroinflammation in astrocytes.	Yes
7	Lin [67]	In vivo	Equol-administered collagen-induced arthritis mice had a lower severity of arthritis symptoms. Equol administration suppressed the expression of IL-6 and its receptor in the inflamed area of collagen-induced arthritis mice.	Yes
8	Yokosuka [34]	In vivo	In ovariectomized mice induced to have intracranial aneurysms, equol protected against aneurysm formation; the disruption of the intestinal microbial conversion of daidzein to equol abolished daidzein’s protective effect against aneurysm formation. Moreover, mice treated with equol had lower inflammatory cytokines in their cerebral arteries.	Yes
9	van der Velpen [35]	Human	In the adipose tissue of postmenopausal women, the expression of inflammation-related genes was upregulated in equol producers but downregulated in non-producers.	Yes
10	Törmälä [68]	Human	ISFs caused a decrease in the VCAM-1 and platelet-selectin. The fall in platelet-selectin was more marked in equol producers. No changes appeared in SHBG, CRP, or ICAM-1.	Yes
11	Reverri [69]	Human	Consuming soy improved arterial stiffness as was assessed by the augmentation index, but did not improve the inflammatory biomarkers (CRP, TNF-α, IL-6, IL-18, and IL-10). The addition of equol-producing status as a covariate did not significantly change these results.	No
12	Nicastro [70]	Human	Equol, while not associated with a decrease in CRP level, was associated with decreased geometric mean WBC counts, comparing the highest quartile to the lowest.	Yes
13	Greany [71]	Human	An RCT of 34 postmenopausal women on 44 mg/day of ISFs showed that the ISFs did not influence the concentrations of Hcy, CRP, sE-selectin, sVCAM-1, and sICAM-1. Equol-producing status did not modify the associations.	No
14	Mangano [72]	Human	In women who received the ISFs intervention, there was no significant differences in percent change in the serum inflammatory markers between equol producers and non-producers.	No

Abbreviations: NO, nitric oxide; PGE2, prostaglandin E_2_; LPS, lipopolysaccharide; INF-γ, interferon gamma; IL-6, interleukin-6; TNF-α, tumor necrosis factor-α; MCP-1, monocyte chemoattractant protein-1; MetS, metabolic syndrome; CRP, C-reactive protein; sICAM, soluble intercellular adhesion molecule; VCAM-1, vascular cell adhesion molecule 1; SHBG, sex hormone-binding globulin; ICAM-1, intercellular adhesion molecule 1; IL-8, interleukin-8; IL-10, interleukin-10; WBC, white blood cell; Hcy, homocysteine; sE-selectin, soluble endothelial leukocyte adhesion molecule-1; TLR4, Toll-like receptor 4; MAPK, mitogen-activated protein kinase; NF-kB, nuclear factor kappa-light-chain-enhancer of activated B cells; RCT, randomized controlled trial.

In vitro studies have consistently shown that equol is anti-inflammatory. First, in macrophages, equol down-regulated the expression of genes related to the production of various kinds of cytokines and inflammatory biomarkers [61] and dose-dependently reduced the secretion of inflammatory biomarkers, such as prostaglandin E2 [61] and MCP-1 [64], whereas daidzein did not have such an effect [61]. In microglia cells, equol reduced the release of interleukin-6 (IL-6) [62,65] and TNF-α [62,65]. These effects may be mediated through the inhibition of Toll-like receptor 4, mitogen-activated protein kinase (MAPK), or NF-κB [65]. In astrocytes, equol attenuated nitric oxide (NO) production and concomitantly decreased the expression of inducible NO synthase [66].

Several in vivo studies in mice reported the effect of equol on inflammatory diseases and inflammation. Equol administration in a rheumatoid arthritis mouse model improved arthritis-induced bone mineral density and suppressed the expression of IL-6 and its receptor in inflamed areas [67]. In ovariectomized mice induced to have intracranial aneurysms, daidzein with equol supplementations protected against aneurysm formation, whereas the disruption of the intestinal microbial conversion of daidzein to equol abolished daidzein’s protective effect against aneurysm formation, indicating that equol alone had a protective effect [34]. The same study also suggested that mice treated with equol had lower inflammatory cytokines in their cerebral arteries [34]. All of the above effects were exerted through the activation of ERβ.

In some of the human RCTs of ISFs, the anti-inflammatory effects of the ISFs were more pronounced in the equol producers than in the non-producers, while in other RCTs, the anti-inflammatory effects of the ISFs were only observed in equol producers. One RCT of ISFs among 110 postmenopausal women on tibolone showed that the equol producers had a larger reduction in platelet-selectin levels than the non-producers after being treated with ISFs [73]. Platelet-selectin is a cell adhesion molecule on the surface of activated ECs [74]. In the peripheral blood mononuclear cells in 30 equol-producing postmenopausal women, ISF intervention down-regulated clusters of genes that were involved in inflammation, oxidative phosphorylation, and cell-cycle regulation, as was suggested by the analysis in the whole-genome gene expression profiles [75]. Further analysis in the same group of participants later reported that ISF supplements upregulated the expression of the anti-inflammatory genes in the adipose tissue of equol producers but down-regulated the expression in non-producers [35].

An RCT of ISFs with 34 women found that neither the ISFs nor the equol-producing status was associated with CRP, VCAM-1, and ICAM-1 [71]. This null finding may be due to the small number of participants.

#### 3.3.2. Antioxidative Effect

Progression of atherosclerotic plaque is caused by molecular changes induced by cytokines and reactive oxidant species (ROS) [76,77]. Myeloperoxidase (MPO) and nicotinamide adenine dinucleotide phosphate (NADPH) oxidase serve as enzymatic sources of oxidant species to generate oxidized LDL (oxLDL) [78,79]. The inhibition of NADPH oxidase activity reduced atherosclerotic lesions in animal models [80,81]. In humans, higher levels of oxidative biomarkers, such as oxLDL, MPO, and F2-isoprostanes, were associated with an increased risk of CHD [82,83,84,85,86,87]. In addition, individuals with MPO deficiency were found to have a very low incidence of CVD [88].

Cell culture studies show that equol is a potent antioxidant in numerous models (Table 2, Figure 4). The equol protected macrophages from oxidative stress induced by L-lactate dehydrogenase or oxLDL by reducing the lipid peroxidation product malondialdehyde (MDA), enhancing the antioxidant glutathione, or increasing the activities of the antioxidant enzyme superoxide dismutase (SOD) [89,90]. Equol treatment on neutrophils that were activated with various stimulants to produce ROS caused reduced toxic action of the ROS; for example, equol decreased the phosphorylation of the proteins regulating NADPH oxidase [91]. Phagocytic cells, after being incubated with equol for 1 h, showed a significant reduction in the intracellular production of superoxide anion and hydrogen peroxide [92]. In the intestinal epithelial cells with oxidative stress induced by hydrogen peroxide, equol promoted the expression of antioxidant genes, increased the activities of SOD, and increased the abundance of nuclear factor erythroid 2 (Nrf2) transcripts [93]. Finally, equol decreased the ratio of reduced/oxidized glutathione in primary cortical neuron cells [94]. Although some antioxidant effects are independent of ERβ [95], these effects are generally considered to be mediated through ERβ.

**Table 2 nutrients-13-03739-t002:** Antioxidative effects of equol.

#	Authors	Type	Findings	Has Effect
1	Lin [93]	In vitro	Equol was shown to protect chicken intestinal epithelial cells from oxidative damage by promoting the expression of antioxidant genes, increasing the activities of antioxidant enzymes, and enhancing antioxidant capacity. Equol significantly enhanced total SOD activity and the Nrf2 transcript.	Yes
2	Pereboom [92]	In vitro	Equol decreased the intracellular production of the superoxide anion and hydrogen peroxide content of phagocytic cells.	Yes
3	Hwang [96]	In vitro	Equol and ascorbic acid interacted synergistically in preventing LDL oxidation. All phases of LDL oxidation were affected by these compounds, which is atypical of the behavior of antioxidants that are consumed during the early phases. Equol was more potent than daidzein and genistein because of its absence of a carbonyl group, C2–C3 double bond and flanking hydroxyl groups in the pyran ring.	Yes
4	Pažoureková [91]	In vitro	Upon activation by ROS, neutrophils treated by equol produced less p40 phox (a component of NADPH oxidase, responsible for the assembly of functional oxidase in intracellular membranes) both extra- and intracellularly to the control.	Yes
5	Choi [97]	In vitro	Equol pretreatment significantly decreased levels of oxidative stress biomarkers such as thiobarbituric acid-reactive substances, carbonyl content and serum 8-hydroxy-2-deoxyguanosine. Moreover, equol increased the activity of CAT, superoxide dismutase, GPx, and glutathione reductase. In addition, equol possessed anticancer activity through acting as an antioxidant and therefore reduced apoptosis.	Yes
6	Wei [98]	In vitro	Low doses of equol could prevent skeletal muscle cell damage induced by hydrogen peroxide. Equol increased cell viability, the concentration of MDA content, and LDH activity.	Yes
7	Kamiyama [99]	In vitro	Equol might contribute to a reduced level of oxLDL-stimulated apoptosis linked to the reduced generation of intracellular ROS in human umbilical vein endothelial cells.	Yes
8	Sierens [100]	In vitro	Equol was able to function as an antioxidant, scavenging potentially harmful free radicals. Equol protected against oxidative-induced DNA damage. Pretreatment of a physiological range of equol offered protection against the hydrogen peroxide-mediated DNA damage in human lymphocytes cells. This protection was greater than that offered by the addition of antioxidant vitamins ascorbic acid and alpha-tocopherol, or the compounds 17β-estradiol and tamoxifen, which have similar structures to ISFs and are known to have moderate antioxidant activity.	Yes
9	Rüfer [101]	In vitro	Equol exhibited higher antioxidant activity than daidzein and about the same antioxidant capacity as the oxidative metabolites of daidzein and genistein despite the lack of the 2,3-double bond with the 4-oxo group and a 5,7-dihydroxyl structure. The antioxidative effect was tested by an ORAC assay which determined the ability of compounds to scavenge peroxyl radicals.	Yes
10	Hwang [51]	In vitro	Equol inhibited LDL oxidation in vitro and LDL oxidative modification by monocyte/macrophages. The antioxidant effect of equol was found to be mediated by the inhibition of superoxide radical production and manifested through enhanced levels of free NO. Equol had a greater antioxidant activity than genistein and daidzein.	Yes
11	Sierens [102]	In vitro	Pretreatment with equol significantly protected sperm DNA against oxidative damage. Compared with ascorbic acid and alpha-tocopherol, being added at physiological concentrations, genistein was the most potent antioxidant, followed by equol, ascorbic acid, and alpha-tocopherol. Equol might have a role to play in antioxidant protection against male infertility.	Yes
12	Arora [103]	In vitro	Compared to genistein and daidzein with their glycosylated and methoxylated derivatives, equol and its 4-hydroxy and 5-hydroxy derivatives were more potent antioxidants, suggesting that the absence of the 2, 3-double bond and the 4-oxo group on the ISF nucleus enhanced antioxidant activity.	Yes
13	Turner [104]	In vitro	Equol inhibited the oxidation of LDL 2.65-fold more than its parent compound daidzein.	Yes
14	Choi [94]	In vitro	Equol acted as an antioxidant in the brains of rats. The ratio of GSH/GSSG in primary cortical neuron cells exposed to equol for 24 and 72 h significantly decreased in a time- and dose-dependent manner. Moreover, equol treatment significantly increased the LDH release in a time-and dose-dependent manner.	Yes
15	Gou [90]	In vitro	Equol protected chicken macrophages from oxidative stress induced by lipopolysaccharide through reducing lipid peroxidation products such as MDA and enhancing the contents of antioxidants such as glutathione and the activities of relevant antioxidase enzymes such as total SOD; effects were also seen in gene expression related to the immune response and increased contents of cytokines.	Yes
16	Liu [105]	In vitro	Equol elevated brain antioxidant activity by increasing SOD, CAT, and GPx levels. MDA levels and AChE activity were decreased in hypertensive and vascular dementia rats. Equol further improved the long- and short-term memory of the rats.	Yes
17	Vedavanam [106]	In vivo	The order of the half-maximal inhibitory concentration values, the indication of the potency of inhibiting glucose-induced LDL lipid peroxidation observed for the compounds, was equol > genistein > daidzein.	Yes
18	Choi [89]	In vivo	Equol might act as an antioxidant through an inhibition of oxidative stress and the stimulation of CAT and SOD, but could also cause pro-oxidant effects, such as the reduction of the GSH/GSSG ratio, depending on the treatment period. A study in mice showed that equol administration significantly inhibited biomarkers of oxidative stress (thiobarbituric acid-reactive substances value, carbonyl content, and serum 8-hydroxydeoxyguanosine). Moreover, the CAT and total SOD activities and their transcripts were significantly increased by equol. Although equol increased the glutathione peroxidase activity in mice treated with equol for 1-week, long-term administration of equol (7 weeks) caused a decrease in the ratio of GSH/GSSG and the activities of GPx and glutathione reductase.	Yes
19	Ma [107]	In vivo	A study in male and ovariectomized female rats with transient middle cerebral artery occlusion revealed that the pretreatment of equol significantly reduced infarct size in both sexes. This neuroprotection was accompanied by a decrease in the NADPH oxidase activity and superoxide levels in the brain. In addition, equol reduced plasma thiobarbituric acid reactive substances and neurological deficits up to 7 days after injury.	Yes
20	Horiuchi [108]	In vitro	The study demonstrated that equol had suppressive effects against oxidative stress in pancreatic β-cells in a dose-dependent manner and presumably through activating PKA signaling.	Yes
21	Jackman [33]	In vivo	Equol exerted weak antioxidant effects in cerebral arteries, whereas the effects of daidzein were insignificant. Antioxidant activity was assessed as the reduction in NADPH-induced superoxide levels.	Yes
22	Widyarini [109]	In vivo	In addition to the activation of estrogenic signaling pathways for photoprotection, equol also provided UV-protective antioxidant effects that depend partially on HO-1 induction. Equol dose-dependently inhibited the oxidative stress measured as UVA-induced lipid peroxidation on mouse skin. A component of the equol lipid protection capacity is attributed to endogenous cutaneous antioxidant enzymes, including the inducible stress protein HO-1.	Yes
23	Nhan [110]	Human	Urinary equol was not associated with the secretion of urinary F2 isoprostane, a measure of cellular lipid peroxidation, after ISF treatment in postmenopausal women. However, the observations on the effect of equol were limited because only two of the eight subjects were equol producers, one of whom experienced a large increase in the biomarker excretion, whereas the other experienced small decreases.	No
24	Hidayat [36]	Human	The level of MDA, an oxidative stress marker, was lower in equol producers than non-producers. This RCT was conducted with 190 postmenopausal women aged 47–60 who received 100 mg ISFs for 6 months. The random allocation of ISFs intervention was carried out separately by equol-producing status.	Yes
25	Richardson [95]	In vitro	Equol might have a beneficial effect in delaying the onset and decreasing the severity of symptoms in Friedreich’s ataxia patients by an antioxidant mechanism, such as reducing the ROS-induced modification of proteins and lipids and impaired mitochondrial function. These effects were independent of the ERβ.	Yes

Abbreviations: ROS, reactive oxygen species; SOD, superoxide dismutase; CAT, catalase; GPx, glutathione peroxidase; MDA, malondialdehyde; AChE, acetylcholinesterase; PKA, protein kinase A; Nrf2, nuclear factor erythroid 2–related factor 2; NADPH, nicotinamide adenine dinucleotide phosphate; LDH, L-lactate dehydrogenase; ORAC, oxygen radical absorbance capacity; GSH/GSSG, reduced/oxidized glutathione; DNA, deoxyribonucleic acid; HO-1, heme oxygenase-1; NQO1, NADPH-quinone oxidoreductase 1; UV, ultraviolet.

Animal studies also demonstrated the antioxidant properties of equol. Two mice studies demonstrated that equol improved vascular health in the brain [33,105]. It was suggested that equol elevated the activities of SOD, catalase, acetylcholinesterase, and glutathione peroxidase, and decreased the MDA levels in mice that had deoxycorticosterone acetate salt-induced hypertension and associated vascular dementia [105]. Equol also reduced NADPH-induced superoxide production in the cerebral arteries of normotensive rats and hypertensive rats that were induced by angiotensin II [33]. However, the antioxidant effect was not observed in daidzein [33]. One study showed that equol suppressed tumor formation in rats, presumably through decreasing the concentrations of thiobarbituric acid-reactive substances and 8-hydroxy-2-deoxyguanosine, and increasing the activity of catalase, SOD, and glutathione peroxidase [97].

One RCT in humans demonstrated that equol-producing individuals could receive the antioxidant benefits from ISFs, but non-producers could not. An RCT of 190 postmenopausal women found that after 6 months of supplementation with ISFs, blood MDA concentrations were significantly lower in the equol producers compared with the non-producers in the ISFs group [36].

At the normal physiological concentrations, equol is a more potent antioxidant compared to ISFs, probably due to the absence of 2,3-double bond, the 4-oxo group, and a 5,7-dihydroxyl [96,101,102,103,106]. The antioxidant property of equol is even more potent than ascorbic acid, alpha-tocopherol, and 17β-estradiol. Comparative studies found that the potency of inhibiting LDL lipid peroxidation was in this order: equol > genistein > daidzein [96,106].

#### 3.3.3. Endothelial Function and Vasodilation

The endothelium regulates vascular tone, carefully balancing vasoconstriction and vasodilation to provide adequate perfusion to target organs [111]. All of the above inflammatory and oxidative processes cause damage to the vascular endothelium, leading to the formation of arteriosclerotic plaques [112]. Flow-mediated dilatation (FMD) of the brachial arteries provides a non-invasive measurement of endothelial function [113].

It has been observed in several categories of vessels (umbilical vein, aorta, pulmonary artery, and cerebral basilar artery) that equol stimulated endothelial redox signaling and increased NO production in the ECs [114,115,116] (Table 3). Studies indicated that equol, by binding to ERβ, could rapidly stimulate the phosphorylation of the extracellular signal-regulated protein kinase 1/2 and phosphatidylinositol 3-kinase/protein kinase B (PI3K/Akt), leading to the activation of endothelial NO synthase (eNOS) [114,117]. NO can react with superoxide anions to form peroxynitrite. NO and peroxynitrite then, in turn, enhance the nuclear accumulation of Nrf2, which binds to an antioxidant response element in target genes to enhance the transcription of the phase II antioxidant defense enzymes, such as superoxide dismutase, catalase, glutathione-S-transferase, glutathione peroxidase, and heme oxygenase-1 [118]. A second pathway by which equol improved endothelial function was through the transactivation of the epidermal growth factor receptor kinase, resulting in a reorganized F-actin cytoskeleton [119]. A third pathway in which equol was involved was through the direct upregulation of eNOS, resulting in reduced oxidative stress in the ECs [115,120]. The target gene eNOS contains an estrogen-response element. Thus, it can be reasoned that the binding of equol to ERβ may be responsible for enhanced eNOS expression. In a fourth pathway, equol could increase the expression of phospho-p38 MAPK and B-cell lymphoma-2 to reduce intracellular ROS production in the ECs [121]. Furthermore, equol directly inhibited the apoptosis of the ECs [122], presumably through stimulating the thymidine incorporation, which is important for the deoxyribonucleic acid (DNA) synthesis of the ECs [122]. This stimulatory effect on cell growth was not observed for daidzein or genistein [122].

**Table 3 nutrients-13-03739-t003:** Endothelial function improvement effects of equol.

#	Authors	Type	Findings	Has Effect
1	Joy [117]	In vitro	Nutritionally relevant plasma concentrations of equol rapidly stimulated phosphorylation of ERK1/2 and PI3K/Akt, leading to the activation of NOS and increased NO production at resting cytosolic Ca^2+^ levels.	Yes
2	Rowlands [119]	In vitro	Equol-stimulated mitochondrial ROS modulated endothelial redox signaling and NO release through transactivation of epidermal growth factor receptor kinase and reorganization of the F-actin cytoskeleton.	Yes
3	Cheng [115]	In vitro	Equol prevented oxidative damage to vascular function in pulmonary cells via downregulating eNOS and oxidative stress.	Yes
4	Zhang [114]	In vitro	In HUVEC, equol increased Nrf2 mRNA as well as the mRNA of the gene products of HO-1 and NQO1. Pretreatment of cells with specific endoplasmic reticulum inhibitors or PI3K/Akt increased Nrf2, HO-1, and NQO1 protein.	Yes
5	Chung [121]	In vitro	Equol had a significant antioxidant effect on the bAECs that were exposed to hydrogen peroxide. Equol pretreatment effectively inhibited the hydrogen peroxide-induced cell death by the reduction in intracellular ROS production, probably through increasing phospho-p38 MAPK.	Yes
6	Zhang [123]	In vitro	The improvement of atherosclerosis by equol through attenuation of endoplasmic reticulum stress is mediated by activating the Nrf2 signaling pathway. Equol treatment inhibited cell apoptosis and attenuated upregulation of endoplasmic reticulum stress markers in HUVECs. In an oxidative stress environment, equol treatment dose-dependently activated the Nrf2 signaling pathway.	Yes
7	Somjen [122]	In vitro	Equol, but not daidzein and genistein, had a monophasic stimulatory effect on thymidine incorporation, which boosts DNA synthesis. In human endothelial cells, equol, daidzein, and genistein stimulated DNA synthesis in a dose-dependent manner. The administration of equol, daidzein, and genistein to immature and ovariectomized female rats resulted in increased creatine phosphokinase in the aorta and in the left ventricle of the heart.	Yes
8	Kim [124]	In vitro	Equol had a vasodilatory effect on human uterine arteries vascular smooth muscle, which was mediated through antagonistic action for a receptor-dependent Ca^2+^ channel.	Yes
9	Johnson [62]	In vitro	Equol exhibited protective effects against NO production in murine microglial cells. Equol also showed greater permeability through artificial gut and blood-brain barriers compared to daidzein.	Yes
10	Chin-Dusting [125]	In vivo	Equol had a dose-dependent inhibition of the contractile responses to noradrenaline in rat isolated aortic rings. Equol independently increased the release of a vasoconstrictor prostanoid, such as thromboxane.	Yes
11	Jackman [33]	In vivo	In normotensive rats, equol displayed vasorelaxant activity similar to daidzein. The relaxant effect of equol was independent of intact endothelium, NOS activity, K^+^ channels, and gender. In the basilar artery, where superoxide levels are higher, equol exerted weak antioxidant effects, whereas the effects of daidzein were insignificant. During hypertension, equol-induced vasorelaxation was preserved, whereas relaxant responses to daidzein were impaired.	Yes
12	Matsumoto [126]	In vivo	Contractions induced by a selective 5-HT receptor agonist increased with insulin treatment, but less so with equol + insulin. In the endothelium-denuded preparations, 5-HT-induced contractions were augmented with insulin treatment but less so by equol + insulin treatment. These differences in 5-HT-induced contractions were eliminated by a large-conductance of Ca^2+^-activated K^+^ channel inhibitor.	Yes
13	Yu [116]	In vivo	Equol significantly increased regional cerebral blood flow in rats and produced an endothelium-independent relaxation in rat cerebral basilar arteries. Selective Ca^2+^-activated K^+^ channel blockers significantly inhibited equol-induced vasodilation in cerebral arteries.	Yes
14	Ohkura [120]	In vivo	Ovariectomized rats were assigned to (1) an ISF-deficient but equol-sufficient group, (2) an ISFs-deficient and equol-deficient group. In the thoracic artery, endothelium-dependent relaxation, cyclic guanosine monophosphate levels in the tissue, and eNOS synthase expression and phosphorylation were significantly higher in the first group compared to the second group.	Yes
15	Törmälä [73]	Human	Before ISF intervention, women with a 4-fold elevation in equol levels had a lower endothelial function index compared to women without this capacity. Soy supplementation had no effect on arterial stiffness or endothelial function in either group.	Yes
16	Kreijkamp-Kaspers [37]	Human	This RCT did not support the hypothesis that ISFs have beneficial effects on endothelial function in older postmenopausal women. However, in the soy-only group, systolic and diastolic blood pressure decreased, and endothelial function improved in the equol producers, whereas blood pressure increased, and endothelial function deteriorated in the non-producers.	Yes
17	Hidayat [36]	Human	ISFs did not improve endothelial functions in both the equol producers and the non-producers. The VCAM-1 and NO did not differ by equol-producing status.	No
18	Clerici [127]	Human	After ISFs treatment, the brachial artery flow-mediated vasodilatation was improved more obviously in the equol producers.	Yes

Abbreviations: ERK1/2, extracellular signal-regulated protein kinases 1 and 2; PI3K/Akt, protein kinase 1/2 and phosphatidylinositol 3-kinase/protein kinase B; NOS, nitric oxide synthase; NO, nitric oxide; ROS, reactive oxygen species; HUVEC, human umbilical vein endothelial cell; eNOS, endothelial nitric oxide synthase; HO-1, heme oxygenase-1; NQO1, NADPH-quinone oxidoreductase 1; Nrf2, nuclear factor-erythroid 2-related factor 2; bAECs, bovine aortic endothelial cell; MAPK, mitogen-activated protein kinase; HUVECs, human umbilical vein endothelial cells; cfPWV, carotid-femoral pulse wave velocity; VCAM-1, vascular cell adhesion molecule-1; 5-HT, 5-hydroxytryptamine; Ca, calcium; K, potassium.

Independent of an effect on the intact endothelium and NOS activity, equol exhibited vasodilator activity in vascular smooth muscle cells of various of arteries, which may be induced by the antagonism of the Ca^2+^-activated K^+^ channel [33,116,124,126]. This vasodilatory effect was not observed with daidzein [33].

Equol-producing status in humans may be critical in unlocking the endothelial benefits of ISFs. In a long-term RCT of ISFs among 202 older postmenopausal women, endothelial function was significantly improved in equol producers, whereas it deteriorated in non-producers when both groups were treated with ISFs [37]. Secondary analysis of the RCT of ISFs with 110 women [73] reported significantly improved endothelial function in equol producers as compared to the controls, whereas the ISFs themselves had no significant effects on endothelial function. Another RCT of ISFs in 62 adults with hypercholesterolemia observed the vasodilatory effect of ISFs on the brachial artery and determined that improvement was larger in equol producers [127].

#### 3.3.4. Arterial Stiffness

A healthy aorta normally exerts a powerful cushioning function, which delivers a nearly steady flow of blood to the end organs [128]. Arterial stiffness impairs this cushioning function and is now recognized as a significant predictor of future cardiovascular events, independent of traditional risk factors [129,130,131]. Arterial stiffness is closely associated with the endothelial function and vasodilation [132] described in the previous section. Arterial stiffness can be determined by measuring carotid-femoral pulse wave velocity (cfPWV), brachial-ankle PWV (baPWV), and the cardio-ankle vascular index (CAVI) [129].

Many nutrients have been hypothesized to improve arterial stiffness, potentially through the modulation of endothelial function, and reduce oxidative stress and inflammatory processes [133]. Pase et al. conducted a systematic review of 38 RCTs of nutrients in 2011 and concluded that supplementation of ISFs and marine-derived omega-3 fatty acids improved arterial stiffness [133]. We recently updated this systematic review of the RCTs of ISFs and conducted a meta-analysis. We showed that supplementation of ISFs significantly improved arterial stiffness [134]. The effect of ISFs on arterial stiffness is independent of blood pressure. However, the duration of intervention in these RCTs was relatively short (23 h to 12 weeks) and the effects of ISFs in these RCTs are likely to result in functional rather than structural changes.

Equol supplements relieve the severity of arterial stiffness in the human body, as is suggested by the RCTs (Table 4). Usui et al. conducted a crossover RCT of 10 mg/day of equol on arterial stiffness assessed by the CAVI in 54 overweight or obese middle-aged men and women in Japan and showed that the supplementation of equol significantly improved arterial stiffness [135]. However, the sample size was small, and the duration of intervention was short (12 weeks). Several sources of evidence support the hypothesis that equol may have a long-term effect on arterial stiffness. A 12-month RCT of ISFs plus epicatechin in 35 patients with diabetes (mean age of 62) showed ISFs plus epicatechin significantly improved arterial stiffness assessed by cfPWV. Although the effect was not solely attributed to the ISFs, the improvement was more pronounced in the equol producers, suggesting that long-term exposure to equol may improve arterial stiffness [136]. Furthermore, the administration of 10 mg/day of equol in 74 women in Japan (mean age of 55) for 12 months significantly reduced arterial stiffness assessed by baPWV [137]. This study was not placebo-controlled; thus, the interpretation of the result is limited. In men prospectively recruited according to equol-producing status, acute (24 h) ISF treatment improved cfPWV in equol producers but had no effect in non-producers [138].

Observational studies in humans also show that equol-producing status or serum equol levels were associated with a lower degree of arterial stiffness. In a cross-sectional study of Japanese women aged 21–89 years, 67 equol-producing women in their 50s showed a significantly lower PWV than 147 non-producers [139]. Although soy had no effect on arterial stiffness measured by augmentation index, women with a 4-fold elevation in equol level cross-sectionally had a lower augmentation index compared to women without the capacity of producing equol [73]. It is worth noting that because the Japanese RCTs comprise a population with the habitual intake of soy ISFs and a high prevalence of equol producers, studies that showed a beneficial effect of ISFs may be attributed to equol. A cross-sectional study of 652 men in Japan (mean age of 49) showed that dietary intake of ISFs was inversely associated with arterial stiffness assessed by baPWV even after adjusting for age and traditional cardiovascular risk factors [140]. Although this study was cross-sectional, the result implies that long-term exposure to equol may improve arterial stiffness. In contrast, in one RCT, equol-producing status did not modify the effect of ISFs on arterial stiffness, measured by peripheral arterial tonometry [69].

#### 3.3.5. Atherosclerosis and CHD

An RCT with mice suggested that equol had anti-atherogenic effects, where equol intervention reduced atherosclerotic lesions in the aorta in apolipoprotein E-deficient mice fed a high-fat diet [123] (Table 5).

Observational studies in humans have shown that equol producers, as compared to non-producers, had significantly lower levels of atherosclerosis. Our cross-sectional study in 272 Japanese men aged 40–49 years found that equol producers had 9-fold lower odds of coronary artery calcification (CAC) presence compared to non-producers, but the serum ISFs were not significantly associated with CAC [142]. The Guangzhou Nutrition and Health Study of 2,572 middle-aged and older adults found that the serum level of equol was prospectively associated with reduced progression of carotid intima-media thickness over 8.8 years [143].

The first and the only observational study that ever explored the association between the equol level and CHD incidence was a nested case-control study within a cohort in China (377 cases and 753 controls). Total urinary ISFs or their metabolites other than equol were not associated with CHD in either women or men. However, urinary equol excretion showed an inverse association with CHD in women. The adjusted hazard ratio (95% confidence intervals) for CHD across increasing quartiles of equol levels were 1.00 (reference), 0.61 (0.32, 1.15), 0.51 (0.26, 0.98), and 0.46 (0.24, 0.89) (*p* = 0.02) [30].

## 4. Discussion

Evidence in this present systematic review suggests that equol has anti-atherogenic properties which are more potent than ISFs. Previous studies investigating the association of the dietary intake of ISFs with CHD have observed controversial results. Our review suggests that the potential protective effects of ISFs on CHD may require the intestinal conversion of daidzein to equol. This systematic review demonstrates that the potential cardioprotective effects of equol are potentially through its anti-inflammatory, vasodilatory, antioxidative properties, as well as its association with the improvement in endothelial functions and arterial stiffness.

In general, cell culture and preclinical studies show that equol is more efficacious than ISFs. Some effects, such as reducing the secretion of inflammatory biomarkers in macrophage [61], reducing NADPH-induced superoxide levels [33], stimulating endothelial cell growth [122], and improving vasodilation [33] were observed with equol treatment only and not with daidzein or genistein. Moreover, the disruption of the intestinal microbial conversion of daidzein to equol abolished daidzein’s protective effect against aneurysm formation [34]. In addition, observational studies showed that equol rather than the ISFs was significantly and inversely associated with CHD incidence [143] as well as the progression of atherosclerosis [142]. In the RCTs of ISFs in humans, the cardioprotective effects of the ISFs were either more pronounced in equol producers than non-producers [73] or were only present in equol producers [35,36]. These observations strongly support the equol hypothesis that equol rather than ISFs are cardioprotective.

Observational studies showed that dietary intake of antioxidants is significantly and inversely associated with the risk of CVD, whereas the RCTs of vitamin E and other antioxidants reported little or no observed risk reduction among participants of a variety of characteristics [144,145,146,147,148,149,150]. One explanation for this discrepancy is that synthetic versions of the antioxidants administered in these trials may not completely mimic the natural forms of the antioxidants [151]. Equol is available as a dietary supplement and can be administered as a natural form because it is developed by the fermentation of soy germ with *Lactococcus* [135].

A large number of studies in this systematic review focused on an estrogen-deprivation situation, e.g., cell models simulating a postmenopausal cellular environment and human studies among postmenopausal women. The effects of estrogen therapy on CVD are controversial. Generally, when blood vessels are healthy, estrogen appears to protect them from the development of atherosclerotic plaques [152]. However, if atherosclerosis is well-established, estrogen does not appear to be beneficial but rather increases the risk of clinical events [153,154]. Because equol has a greater affinity for ERβ than for ERα, equol may not have adverse effects associated with estrogen supplementation, which is considered to target ERα [155].

Overall, we thoroughly reviewed the potential protective effects of equol on CHD from a comprehensive collection of studies, including transcriptomics, whole-genome gene expression, in vitro, animal, and observational and clinical studies. Based on this systematic review, the clinical studies of equol on atherosclerosis and CHD are warranted.

## Figures and Tables

**Figure 1 nutrients-13-03739-f001:**
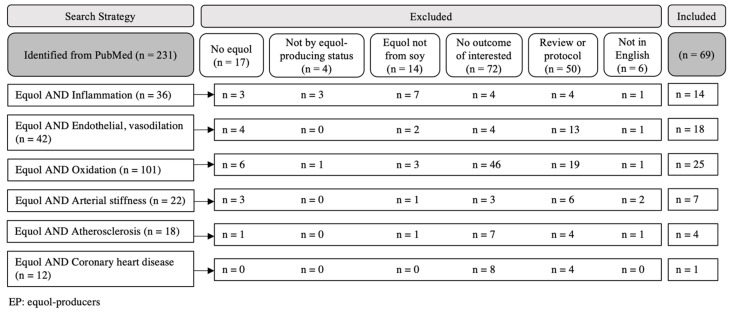
Search strategies in PubMed (inception to 28 July 2021) and the literature selection process.

**Figure 2 nutrients-13-03739-f002:**
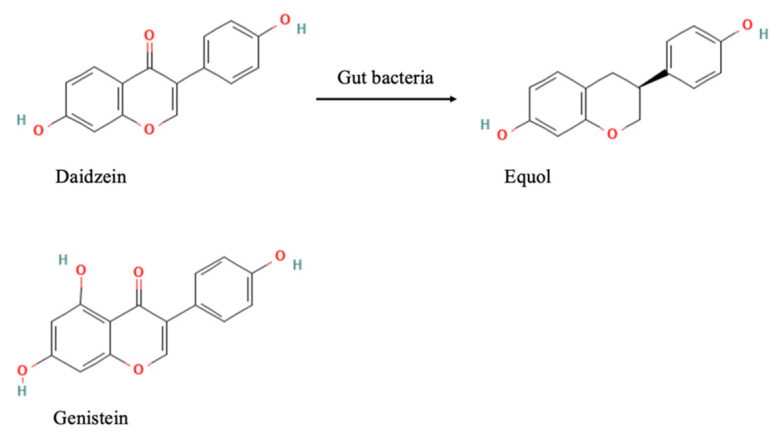
Structures of genistein, daidzein, and equol [48,49,50]. Genistein and daidzein are two major ISFs and comprise >95% of dietary sources. Equol is a metabolite of daidzein, biotransformed by gut bacteria.

**Figure 3 nutrients-13-03739-f003:**
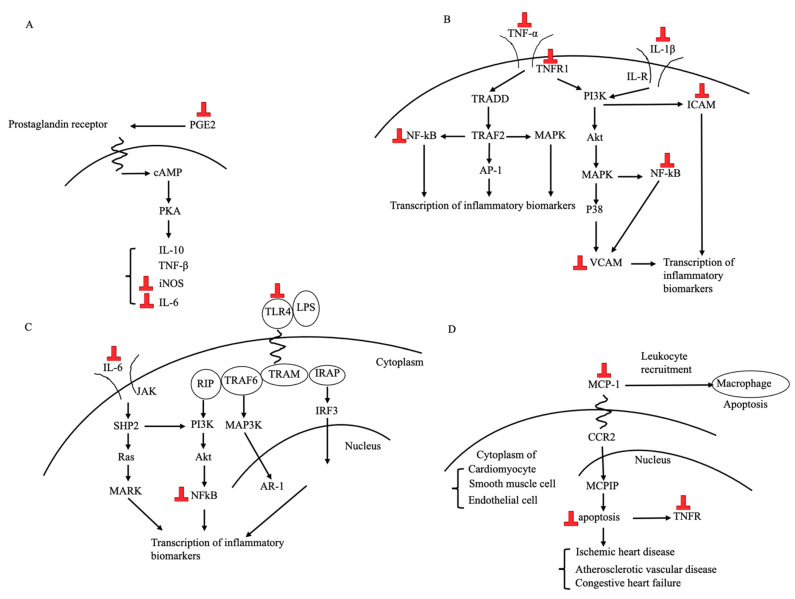
Signaling and pathways in which equol exerts anti-inflammatory effects. (**A**): PGE2 pathway [61]. (**B**): TNF-α [62] and IL-1 pathway [69]. (**C**): TLR4 [65] and IL-6 pathway [69]. (**D**): MCP-1 pathway [64]. “⏊” indicates the inhibitory effect by equol. PGE2: prostaglandin E2, cAMP: cyclic adenosine monophosphate, PKA: protein kinase A, IL: interleukin, TNF: tumor necrosis factor, iNOS: nitric oxide system, TNFR: TNF receptor, TRADD: TNFR1-associated death domain protein, TRAF2: TNF receptor-associated factor 2, PI3K or Akt: phosphoinositide 3-kinases, MAPK or P38: mitogen-activated protein kinase, AP: activator protein, NF-kB: nuclear factor kappa B, ICAM: intercellular adhesion molecule, VCAM: vascular cell adhesion molecule, IL-R: interleukin receptor, LPS: lipopolysaccharide, TLR: Toll-like receptor, TRIF: Toll/IL-1R domain-containing adaptor-inducing IFN-β, TRAM: TRIF-related adaptor molecule, TRAF: TNFR-associated factor, RIP: receptor-interacting protein, IRF: IFN regulatory factor, IFN: interferon, JAK: janus kinase, SHP: Src homology-2 domain-containing protein tyrosine phosphatase, AR: androgen receptor, MCP: monocyte chemoattractant protein, CCR: Chemokine receptor, MCPIP: monocyte chemoattractant protein-induced protein.

**Figure 4 nutrients-13-03739-f004:**
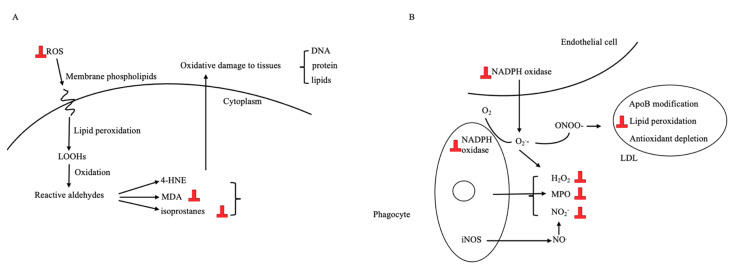
Signaling and pathways on which equol exerts antioxidative effect. (**A**): ROS pathway [99]. (**B**): NADPH pathway [91]. “⏊” indicates the inhibitory effect by equol. MDA: malondialdehyde, 4-HNE: dyhidroxynonel, LOOHs: lipid hydroperoxides, ROS: reactive oxygen, DNA: deoxyribonucleic acid, ONOO-: peroxynitrite, NO_2_: nitrogen dioxide, O_2_-: superoxide, MPO: myeloperoxidase, NADPH: nicotinamide adenine dinucleotide phosphate, apoB: apolipoprotein B, iNOS: inducible nitric oxide synthase, H_2_O_2_: hydrogen peroxide.

**Table 4 nutrients-13-03739-t004:** Arterial stiffness preventive effects of equol.

#	Authors	Type	Findings	Has Effect
1	Usui [135]	Human	Compared with the placebo group, intervention with natural equol led to a significant decrease in the HbA1c, serum LDL-C levels and CAVI score. Furthermore, the effect was more prominent in a subgroup of female equol non-producers.	Yes
2	Curtis [136]	Human	Overall, the ISF intervention did not significantly change the common carotid artery or augmentation index, but the pulse pressure variability improved. Equol producers had larger reductions in mean arterial pressure and PWV compared with non–producers.	Yes
3	Hazim [138]	Human	In an RCT, acute ISF treatment (24 h) improved cfPWV in equol producers but had no effect on endothelial function and NO in non-producers.	Yes
4	Yoshikata [137]	Human	Reduction in arterial stiffness was observed after 12 months of equol supplementation. Significant reductions in respective parameters were observed in women with moderate to high risk for arteriosclerosis, hypertriglyceridemia, bone resorption risk, and bone fracture risk.	Yes
5	Yoshikata [139]	Human	Equol-producing women in their 50 s showed significantly lower PWV. In a multivariate logistic regression, for women in their 50 s, equol production was significantly associated with lower arterial stiffness.	Yes
6	Reverri [69]	Human	Consuming soy nuts improved arterial stiffness as assessed by the augmentation index using peripheral arterial tonometry. Addition of equol-producing status as a covariate did not significantly change the result.	No
7	Törmälä [73]	Human	Soy supplementation had no effect on arterial stiffness in either equol producers or non-producers. At baseline (before ISF treatment), women with a 4-fold elevation in equol level had a lower augmentation index compared to women without this capacity.	Yes

Abbreviations: LDL-C, low-density lipoprotein cholesterol; CAVI, cardio-ankle vascular index; cfPWV, carotid-femoral pulse wave velocity; HbA1c, glycated haemoglobin; ISF, isoflavones; PWV, pulse wave velocity; NO, nitric oxide.

**Table 5 nutrients-13-03739-t005:** Anti-atherosclerotic and CHD preventive effects of equol.

#	Authors	Type	Findings	Has Effect
1	Eyster [141]	In vivo	Equol did not impact atherosclerotic lesions. Similar responses of genes to both equol and estradiol might reflect that equol served as a natural selective estrogen receptor modulator in the arteries. Equol modulated the expression of 10 genes in the atherosclerosis model that estradiol did not.	No
2	Zhang [123]	In vivo	Equol intervention reduced atherosclerotic lesions in the aorta in high-fat-diet treated apolipoprotein E-deficient mice. Plasma lipid analysis showed that equol intervention reduced triglycerides, TC, and LDL-C and increased HDL-C.	Yes
3	Ahuja [142]	Human	In multivariable models, the odds ratio for the presence of CAC in equol producers compared with the equol in non-producers was 0.10 (95 % confidence interval: 0.01, 0.90, *p* < 0.04). However, serum ISFs were not related to CAC.	Yes
4	Zuo [143]	Human	An 8.8-year prospective study including 2572 subjects (40 to 75 years old) found that ISFs and equol were associated with reduced progression of carotid intima-media thickness. Path analyses indicated that the association of serum equol with atherosclerosis was mediated by increased SHBG and decreased blood pressure but not lipids.	Yes
5	Zhang [30]	Human	Urinary levels of ISFs and other metabolites of ISFs were not associated with incident CHD, while urinary equol was significantly associated with CHD. The adjusted odds ratios (95% confidence intervals) for CHD across increasing quartiles of equol levels in women were 1 (reference), 0.61 (0.32, 1.15), 0.51 (0.26, 0.98), and 0.46 (0.24, 0.89) (*p* = 0.02 for trend).	Yes

Abbreviations: apoE, apolipoprotein E; cIMT, carotid intima–media thickness; HDL, high-density lipoprotein; LDL, low-density lipoprotein; SBG, systolic blood pressure; SHBG, sex hormone-binding globulin; TC, total cholesterol; CAC, coronary artery calcium; SHBG, sex hormone-binding globulin; CHD, coronary heart disease.

## Data Availability

Not applicable.

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
