# Peer review of "Potential Protective Effects of Equol (Soy Isoflavone Metabolite) on Coronary Heart Diseases—From Molecular Mechanisms to Studies in Humans"

_nutrients, 2021, doi:10.3390/nu13113739_

Round 1
Reviewer 1 Report
The authors summarized the evidence that equol, a soy isoflavone metabolite from gut microbiome, has anti-oxidative, ant-inflammatory effects and improve arterial stiffness and atherosclerosis, and equol have a greater cardioprotective benefit than soy isoflavones. This study suggested equol, as dietary supplement, may be beneficial to coronary heart diseases. Even though equol has known to active ERβ, how equol, not daidzein or genistein, has these protective effect remain unclear. Many effects are generally considered to be mediated through ERβ, but whether this is directly regulated or indirectly by ERβ remain unclear. Besides, does any study indicate that equol is beneficial to all age of human or specific age? How does equol have cardioprotective roles, or what are mechanisms in the heart tissue?
Reviewer 2 Report
The present study describes the protective effects of a soy isoflavone metabolite equol on coronary heart disease. Authors summarized anti-inflammatory, antioxidant, endothelial function improvement, arterial stiffness inhibitory, as well as anti-atherosclerotic and CHD effects of equol and clearly presented the findings in tabular format. They have done a great job by depicting the molecular mechanisms underlying the anti-inflammatory and antioxidant activities of equol. It would give readers a clear idea about the protective effects of equol.
Author Response
Response to Reviewer 2 Comments
Point 1: The present study describes the protective effects of a soy isoflavone metabolite equol on coronary heart disease. Authors summarized anti-inflammatory, antioxidant, endothelial function improvement, arterial stiffness inhibitory, as well as anti-atherosclerotic and CHD effects of equol and clearly presented the findings in tabular format. They have done a great job by depicting the molecular mechanisms underlying the anti-inflammatory and antioxidant activities of equol. It would give readers a clear idea about the protective effects of equol.
Response 1: Thank you for the kind evaluation and for recognizing the value of our paper!
Reviewer 3 Report
The current review described the potential anti-inflammatory, vasodilatory, anti-oxidative, and cardio protective effects of equol in details. Authors suggested that equol intake could improve endothelial functions and arterial stiffness. Overall, the manuscript is well written and has all the merit to publish in this journal. There are few minor comments as follows;
Biosynthesis of equol from daidzein is a crucial step. Authors may discuss gut micro-organisms and enzymes involved in the conversion more elaborately.
Some of the reported studies “Greany et al., (Ref-65); Mangano et al., (Ref-68)” did not find any anti-inflammatory effects of equol. Is there any role of “Estrogen”? How authors can explain these results?
Authors reported more potent anti-oxidant effects of equol compared to other isoflavones. What is the mechanism of this higher activity? Is there any report of combination therapy showing synergistic activity with equol?
